

# A revision of *Plectanocotyle* (Monogenea, Plectanocotylidae), with molecular barcoding of three species and the description of a new species from the streaked gurnard *Chelidonichthys lastoviza* off Algeria

Zouhour El Mouna Ayadi[1], Fadila Tazerouti[1], Delphine Gey[2] and Jean-Lou Justine[3]

[1] Faculté des Sciences Biologiques, Université des Sciences et de la Technologie Houari Boumediene, Alger, Algeria
[2] Molécules de Communication et Adaptation des Micro-Organismes, Muséum National d'Histoire Naturelle, Paris, France
[3] ISYEB-Institut de Systématique, Évolution, Biodiversité, Muséum National d'Histoire Naturelle, Paris, France

Corresponding author
Jean-Lou Justine, justine@mnhn.fr

## ABSTRACT

**Background:** The family Plectanocotylidae includes parasites of the gills of marine fish; although nine genera and about 20 species have been described, almost no molecular information is available. Putting aside *Plectanocotyle elliptica* Diesing, 1850, supposedly a parasite of the white perch *Morone americana*, never found again since its original description, two species were valid within *Plectanocotyle* Diesing, 1850 before this work: *Plectanocotyle gurnardi* (Van Beneden & Hesse, 1863) Llewellyn, 1941 and *Plectanocotyle major* Boudaya, Neifar & Euzet, 2006.
**Methods:** In this paper, we describe the third species of the genus *Plectanocotyle* and perform a comparative morphological and molecular analysis of the three species and of *Triglicola obscura* (Euzet & Suriano, 1974) Mamaev, 1976. Host fishes were also barcoded (COI) for confirmation of host identifications.
**Results:** *Plectanocotyle lastovizae* n. sp. is described from the gills of the streaked gurnard *Chelidonichthys lastoviza* collected off Algeria. The species is compared with specimens of *Plectanocotyle* cf. *gurnardi* (from *C. lastoviza*) from the same locality and *P. major* and *T. obscura* (both from the longfin gurnard *C. obscurus*). Molecules from *Plectanocotyle* cf. *gurnardi* could not be compared with *P. gurnardi* from the type-host and type-locality and we kept the status of the Mediterranean specimens as pending. Algeria is a new geographic record for *P. major* and *T. obscura*. *Plectanocotyle lastovizae* n. sp. is distinguished from the other species found in the Mediterranean by the measurements of clamps, number of testes, and COI sequences, with notable divergence (7.8–11.8%) from the other two species of the genus.
**Discussion:** We briefly present a list of currently known members of the family Plectanocotylidae, their biology and their hosts.

## INTRODUCTION

The Plectanocotylidae Monticelli, 1903 are a family of polyopisthocotylean monogeneans which are distinguished by the structure of the clamps, having a large unpaired accessory piece (*Yamaguti, 1963*). In addition, all plectanocotylids have a fixed number of clamps (6 or 8) and their male copulatory organ consists of a sheaf of long spines. The Plectanocotylidae currently include nine genera and all species are parasites of marine teleost fishes (*WoRMS, 2021*). Two species are currently valid in the genus *Plectanocotyle* Diesing, 1850: *Plectanocotyle gurnardi* (Van Beneden & Hesse, 1863) Llewellyn, 1941 and *Plectanocotyle major* Boudaya, Neifar & Euzet, 2006 (*Boudaya, Neifar & Euzet, 2006*; *Llewellyn, 1941*; *Van Beneden & Hesse, 1863*); both are parasites on the gills of triglid fish (gurnards). The type-species of the genus, *Plectanocotyle elliptica* Diesing, 1850, was described from the white perch *Morone americana* (Gmelin, 1789), a member of the Moronidae, but was not found again since its original description. We consider here that only the two species mentioned above were valid before this work; the status of *P. elliptica* is discussed later in this article.

In this paper, we report the finding of a third species of the genus, found off Algeria on the gills of the streaked gurnard *Chelidonichthys lastoviza*. We compared the morphology of our new species to two other species, collected at the same place, and we performed a molecular analysis of three species of the genus and of *Triglicola obscura* (Euzet & Suriano, 1974) Mamaev, 1976, a species also found on one of the hosts studied here. We collected specimens of a species which could be attributed to *P. gurnardi*; however, its host was different from the type-host (*C. lastoviza* vs *Eutrigla gurnardus*) and the locality was different from the type-locality (Mediterranean vs North Sea); therefore, we designate this species as *Plectanocotyle* cf. *gurnardi* throughout this study but do not discuss it further in the absence of molecular information from *P. gurnardi* from its type-host and type-locality.

The host fish were also barcoded, and we applied rigorous methods to ensure traceability of both fish and monogenean specimens and adequate barcoding (COI) of both hosts and parasites.

We also briefly discuss a list of currently known Plectanocotylidae, their biology and their hosts. This paper is part of a series on monogeneans of fishes of the South shore of the Mediterranean Sea (*i.e.*, *Ayadi et al., 2017*; *Azizi et al., 2021*; *Boudaya & Neifar, 2016*; *Bouguerche et al., 2019a*, *2019b*, *2019c*; *Bouguerche, Justine & Tazerouti, 2020*; *Bouguerche et al., 2020*, *2021*; *Chaabane, Neifar & Justine, 2015*; *Chaabane et al., 2016a*, *2016b*; *Chaabane, Neifar & Justine, 2017*; *Kheddam, Justine & Tazerouti, 2016*, *2020*).

## MATERIALS AND METHODS

### Fish

Gurnards, fifty *Chelidonichthys lastoviza* (Bonnaterre, 1788) and eighty *Chelidonichthys obscurus* (Walbaum, 1792) were collected from Bouharoun, Algerian

coast (36°37′24.17″N, 2°39′17.38″E). Fish were dead when purchased. Fish specimens were identified using keys (*Fischer, Bauchot & Schneider, 1987*) and transferred to the laboratory shortly after capture. Gills were removed carefully from each fish and observed under a microscope for the presence of monogeneans.

## Monogeneans

Monogeneans were removed alive from gills, dehydrated, stained with acetic carmine and mounted in Canada balsam according to routine methods (*Ayadi et al., 2017*). Some specimens were examined in Berlese fluid. Drawings were made with the help of an Olympus BH-2 microscope drawing tube, then scanned and redrawn on a computer with Adobe Illustrator. Measurements are in micrometres.

## Traceability of fish, monogenean specimens and host-parasite relationships

For the molecular study, we ensured that hosts and monogeneans were labelled with respect to host-parasites relationships, *i.e.*, complete traceability. A tissue sample of the fish was taken and several monogeneans were extracted; each monogenean was cut in two halves, the posterior half being processed for molecular analysis and the anterior being kept for morphological assessment and preparation of a voucher slide. This ensures that the molecular identification of the host fish and their monogenean parasites correspond perfectly (*Ayadi et al., 2017*; *Azizi et al., 2021*; *Bouguerche et al., 2019a*, *2019b*, *2019c*, *2020*, *2021*; *Justine et al., 2013*), at the individual fish and parasite level (Table 1). Slides were deposited in the Muséum National d'Histoire Naturelle, Paris, France (MNHN), under registration numbers MNHN HEL1652-HEL1750.

## Molecular barcoding of fish

Total genomic DNA was isolated using a QIAamp DNA Mini Kit (Qiagen), as per the manufacturer's instructions. The 5′ region of the mitochondrial cytochrome c oxidase subunit I (COI) gene was amplified with the primers FishF1 and FishR1 (*Ward et al., 2005*). PCR reactions and amplification were performed as in *Ayadi et al. (2017)*. We used CodonCode Aligner version 3.7.1 software (CodonCode Corporation, Dedham, MA, USA) to edit sequences, which were 655 bp in length, compared them to the GenBank database content with BLAST, and deposited them in GenBank under accession numbers MG761757–MG761759 and MW788679–MW788687 (Table 1). Species identification was confirmed with the BOLD identification engine (*Ratnasingham & Hebert, 2007*).

## COI sequences of monogeneans

Total genomic DNA was isolated using a QIAmp DNA Micro Kit (Qiagen). The specific primers JB3 (= COI-ASmit1) and JB4.5 (= COI-ASmit2) were used to amplify a fragment of 424 bp of the COI gene (*Bowles, Blair & McManus, 1995*; *Littlewood, Rohde & Clough, 1997*). PCR reactions and amplification were performed as in *Ayadi et al. (2017)*. Sequences were edited with CodonCode Aligner 3.7.1, compared to the GenBank database content with BLAST, and deposited in GenBank under accession numbers MG761760–MG761766 and MW796584–MW796594.

**Table 1 Fish and monogeneans, and their morphological and molecular identifications.**

| Fish id | Fish morphological identification | Fish COI GenBank # | Fish molecular identification (BOLD) | Monogenean id | Monogenean morphological identification | Monogenean COI GenBank # | Monogenean slides deposited |
|---|---|---|---|---|---|---|---|
| Br23 | *Chelidonichthys lastoviza* | MG761757 | *Chelidonichthys lastoviza* 99.84% | Br23Mo1 | *Plectanocotyle lastovizae* n. sp. | MG761760 | HEL1723 |
| | | | | Br23Mo2 | *Plectanocotyle lastovizae* n. sp. | MG761761 | HEL1724 |
| | | | | Br23Mo3 | *Plectanocotyle lastovizae* n. sp. | MG761762 | HEL1725 |
| Br26 | *Chelidonichthys lastoviza* | MG761758 | *Chelidonichthys lastoviza* 100% | Br26Mo1 | *Plectanocotyle* cf. *gurnardi* | MG761763 | HEL1727 |
| | | | | Br26Mo3 | *Plectanocotyle* cf. *gurnardi* | MG761764 | HEL1728 |
| Br29 | *Chelidonichthys obscurus* | MG761759 | *Chelidonichthys obscurus* 99.69% | Br29Mo1 | *Triglicola obscura* | MG761765 | HEL1738 |
| | | | | Br29Mo2 | *Triglicola obscura* | MG761766 | HEL1739 |
| | | | | Br29Mo3 | *Triglicola obscura* | MW796584 | HEL1740 |
| Br 30 | *Chelidonichthys lastoviza* | MW788679 | *Chelidonichthys lastoviza* 100% | Br30Mo1 | *Plectanocotyle lastovizae* n. sp. | MW796585 | HEL1726 |
| Br 31 | *Chelidonichthys lastoviza* | MW788680 | *Chelidonichthys lastoviza* 100% | Br31Mo3 | *Plectanocotyle* cf. *gurnardi* | MW796586 | HEL1729 |
| | | | | Br31Mo4 | *Plectanocotyle* cf. *gurnardi* | MW796587 | HEL1730 |
| | | | | Br31Mo5 | *Plectanocotyle* cf. *gurnardi* | MW796588 | HEL1731 |
| Br 32 | *Chelidonichthys lastoviza* | MW788681 | *Chelidonichthys lastoviza* 100% | Br32Mo1 | *Plectanocotyle* cf. *gurnardi* | MW796589 | HEL1732 |
| | | | | Br32Mo2 | *Plectanocotyle* cf. *gurnardi* | MW796590 | HEL1733 |
| | | | | Br32Mo3 | *Plectanocotyle* cf. *gurnardi* | MW796591 | HEL1734 |
| Br 33 | *Chelidonichthys obscurus* | MW788682 | *Chelidonichthys obscurus* 99.85% | Br33Mo1 | *Plectanocotyle major* | MW796592 | HEL1735 |
| Br 36 | *Chelidonichthys obscurus* | – | – | Br36Mo1 | *Plectanocotyle major* | MW796593 | HEL1736 |
| Br 37 | *Chelidonichthys obscurus* | MW788685 | *Chelidonichthys obscurus* 100% | Br37Mo1 | *Plectanocotyle major* | MW796594 | HEL1737 |

**Note:**

Note that traceability was ensured by labelling and barcoding fish individuals and their respective monogenean parasites. New data include eight new sequences of fish and 18 new sequences of monogeneans. In addition, some fish were barcoded but monogenean sequences from the same fish were not retrieved. These were registered in GenBank as: MW788683: Br34 *Chelidonichthys obscurus* 100%; MW788684: Br35 *Chelidonichthys obscurus* 99.85%; MW788686: Br38 *Chelidonichthys obscurus* 100%; and MW788687: Br39 *Chelidonichthys obscurus* 99.69% (percentages are BOLD similarity). Some monogeneans with voucher slides did not provide molecular sequences (slides HEL1741–1750).

## Trees and distances

Two matrices were constructed, for delimitation of taxa and phylogeny.

The first matrix was built from all our new sequences and the few COI sequences of plectanocotylids already in GenBank. In particular, we used COI sequences from plectanocotylids collected off Sète, on the Mediterranean coast of France (*Jovelin & Justine,*

*2001*). Since the Family Mazocraeidae is considered close to the Plectanocotylidae in phylogenies (*Olson & Littlewood, 2002*), we used a sequence of a mazocraeid as outgroup; the COI sequence with the best coverage with our sequences was found within the complete mitogenome of *Neomazocraes dorosomatis* (Yamaguti, 1938) Price, 1943 (JQ038229, unpublished sequence by Zhang, Wu, Xie & Li, 2018). On this matrix with 21 sequences, after estimating the best model with MEGA7 (*Kumar, Stecher & Tamura, 2016*), a tree was inferred using the Maximum Likelihood method based on the Hasegawa-Kishino-Yano model with Gamma distribution (HKY+G) in MEGA7, with 1,000 bootstrap replications (*Hasegawa, Kishino & Yano, 1985*). The sequences were 396 bases in length, but the analysis was on only 312 positions because of some indels and incomplete coverages of some sequences.

A second matrix was built from the first by deleting all sequences with indels or missing parts. The second matrix had only 15 sequences, but all taxa of the first matrix were represented. All sequences were perfectly clean: there were 396 positions in the dataset for sequences which were 396 bases in length. After estimating the best model with MEGA7, a tree was inferred using the Maximum Likelihood method based on the Hasegawa-Kishino-Yano model with Gamma distribution (HKY+G) in MEGA7, with 1,000 bootstrap replications (*Hasegawa, Kishino & Yano, 1985*). A tree was also inferred with the Neighbour-Joining method with MEGA7 with 1,000 bootstrap replications.

Distances were computed on MEGA7 on the second "clean" matrix, using the MEGA7 "group" feature for convenience (groups were defined on the basis of clades found in the analysis). Kimura-2 and *p*-distances were computed, for comparison with other studies.

## Nomenclature

The electronic version of this article in Portable Document Format (PDF) will represent a published work according to the International Commission on Zoological Nomenclature (ICZN), and hence the new names contained in the electronic version are effectively published under that Code from the electronic edition alone. This published work and the nomenclatural acts it contains have been registered in ZooBank, the online registration system for the ICZN. The ZooBank LSIDs (Life Science Identifiers) can be resolved and the associated information viewed through any standard web browser by appending the LSID to the prefix http://zoobank.org/. The LSID for this publication is: urn:lsid: zoobank.org:pub:B6FB3B21-138D-49AD-B0D6-456AC3F7D36E. The online version of this work is archived and available from the following digital repositories: PeerJ, PubMed Central and CLOCKSS.

## RESULTS

### Molecular identification of fish

The identification of fish species based on morphology was confirmed by a DNA barcoding approach. BLAST analysis of the COI sequences with the NCBI and BOLD databases showed sequence similarity values close to 100%, thus confirming the morphological identifications (Table 1).

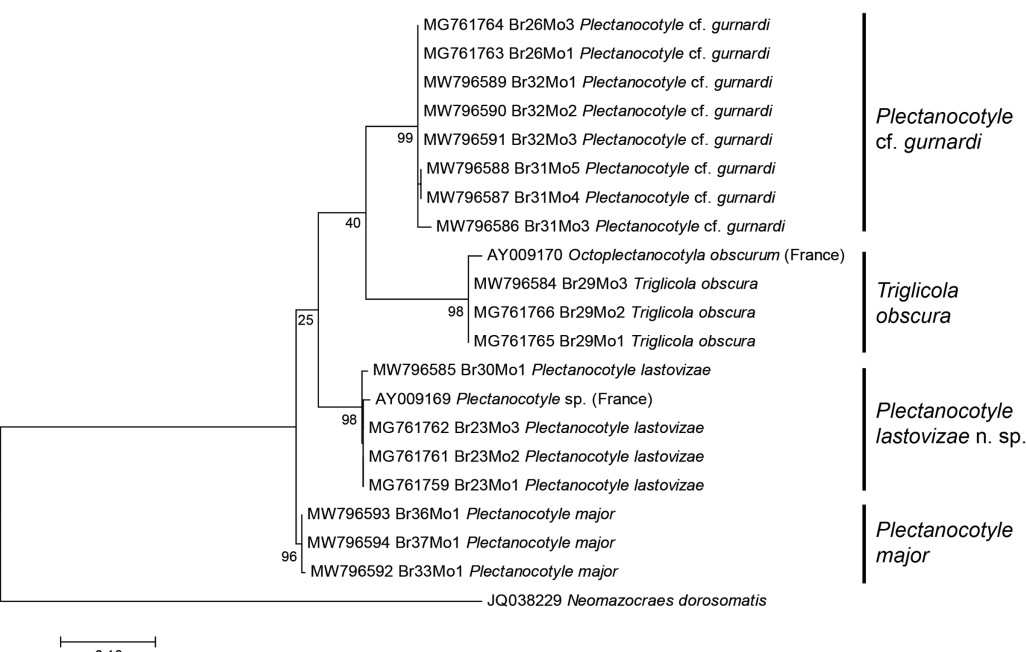

**Figure 1 Molecular phylogenetic analysis of all available members of the Plectanocotylidae.** The tree was obtained by the ML method from COI sequences. All available sequences were used (18 new sequences + 3 from GenBank). Each species is represented by a clade with high bootstrap value. The two sequences from monogeneans collected in France (*Jovelin & Justine, 2001*) are assigned each to a clade: "*Plectanocotyle* sp." to the *P. lastovizae* clade, and "*Octoplectanocotyla obscurum*" to the *Triglicola obscura* clade. Bootstrap values for nodes higher than species are low; *Triglicola obscura* is not a sister-group to *Plectanocotyle* in this tree. Scale: base differences per site.

## Molecular characterization of monogeneans

A tree built from the first matrix, which included all available COI sequences of Plectanocotylidae (our 18 new sequences and 2 already in GenBank) and 1 Mazocraeidae, provided the following results (Fig. 1). The analysis involved 21 nucleotide sequences, and there were a total of 312 positions in the final dataset. The purpose of this matrix with all sequences was to see whether it was possible to attribute the available sequences to separate taxa, and to verify the identity of sequences from GenBank. The answer was positive; the ML tree showed several clades with 100% bootstrap support and each clade corresponded to a nominal species identified from morphology. The *Plectanocotyle* cf. *gurnardi* clade included 8 sequences, all new. The *P. major* clade included three sequences, all new. The *P. lastovizae* clade included three new sequences and the sequence labelled "*Plectanocotyle* sp" in GenBank (AY009169). The *T. obscura* clade included three new sequences and the sequence labelled "*Octoplectanocotyla obscurum*" in GenBank (AY009170). Although the analysis provided high bootstrap values for species nodes, the higher nodes had very low values, and *Triglicola* was not a sister-group to *Plectanocotyle*. We considered that low bootstrap values were caused by the presence of many indels in some sequences.

The second matrix was built by excluding from the first matrix all sequences with indels or incomplete coverage, to improve the accuracy of the final analysis; the matrix, including
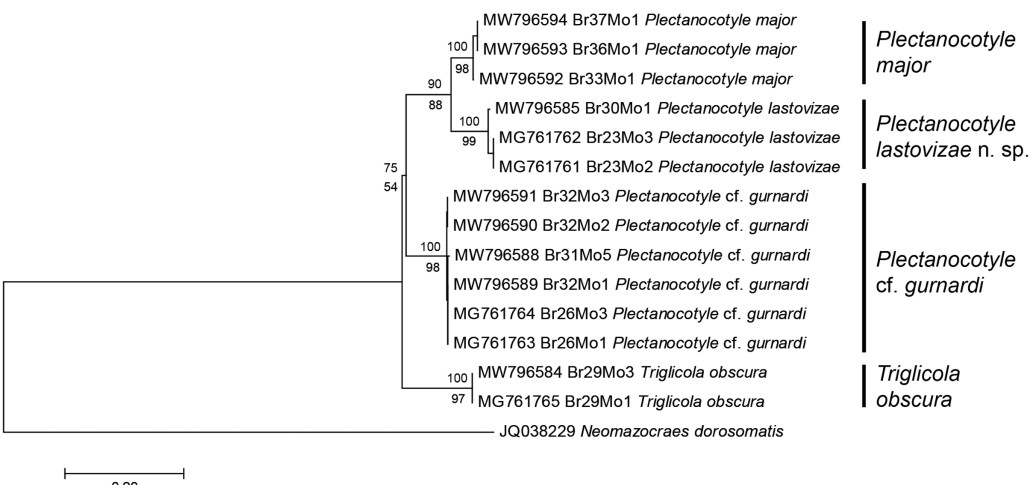

**Figure 2 Molecular phylogenetic analysis of selected members of the Plectanocotylidae.** Only clean COI sequences were selected and the dataset had 396 positions for sequences 396 base pairs in length. The tree was obtained by the Maximum Likelihood method and a similar tree was obtained by the NJ method; bootstrap values for ML below branches, for NJ above branches. Each nominal species is represented by a distinct clade with high bootstrap values (97–99 ML; 100 NJ). *Plectanocotyle lastovizae* is sister-group to *P. major* (90 ML; 88 NJ). A sister-group relationship between *Triglicola obscura* and a clade formed by the three *Plectanocotyle* species was found, but with low values (54 ML; 75 NJ). Scale: base differences per site.

15 sequences, was perfectly clean, with 396 positions for 396 base-long sequences. Excluded sequences were the old sequences from 2001 (*Jovelin & Justine, 2001*) and a few of our new sequences. The ML tree built from the second matrix (Fig. 2) showed the same four plectanocotylid taxa, with *T. obscura* as a sister-group to a clade including the three *Plectanocotyle* species. The NJ tree based on the same matrix had an identical topology. As in the first analysis, support for each species was high (98–99 ML, 100 NJ). Supports for higher nodes were higher than in the first analysis, albeit still relatively low. The support for the *Plectanocotyle* clade (sister-group to *Triglicola*) was low in ML (54) but relatively high in NJ (75). Within this clade, *P.* cf. *gurnardi* was the sister-group to a clade (88ML, 90 NJ) including the two sister-groups *P. lastovizae* and *P. major*.

Distances were computed on the basis of the second matrix (Table 2). Variation was low within each species (*P.* cf. *gurnardi*, 0.220% and 0.221%; *P. major*, 0.509% and 0.512%; *P. lastovizae*, 0.678% and 0.684%, respectively for p-distance and Kimura-2 distances). The three sequences of *T. obscura* were identical (variation 0%). Distances between species were high, at 7.80% and 8.31% between *P. lastovizae* and *P. major*, and at 12.55% and 13.82% between *P. major* and *T. obscura* (each time for p-distance and Kimura-2 distances, respectively)

These results strongly suggest that the three species found in Algerian marine waters, *Plectanocotyle lastovizae* n. sp., *Plectanocotyle major* and *Plectanocotyle* cf. *gurnardi*, are distinct at the molecular level, with interspecific distances ranging between 7.8 and 11.8%. Molecular results justify description of a new species, in addition to the morphological differences between species.

**Table 2 Genetic distances between COI sequences of Monogeneans.**

| p-distances | P. lastovizae | P. major | P. cf. gurnardi | T. obscura |
|---|---|---|---|---|
| *Plectanocotyle lastovizae* | *0.678* | | | |
| *Plectanocotyle major* | 7.80 | *0.509* | | |
| *Plectanocotyle* cf. *gurnardi* | 10.90 | 10.30 | *0.220* | |
| *Triglicola obscura* | 11.87 | 12.55 | 11.75 | *0.000* |
| Kimura-2 distances | P. lastovizae | P. major | P. cf. gurnardi | T. obscura |
| *Plectanocotyle lastovizae* | *0.684* | | | |
| *Plectanocotyle major* | 8.31 | *0.512* | | |
| *Plectanocotyle* cf. *gurnardi* | 11.83 | 11.11 | *0.221* | |
| *Triglicola obscura* | 13.00 | 13.82 | 12.99 | *0.000* |

Note:
Distances are percentages based on the matrix including only clean sequences. Kimura-2 and p-distances are indicated. Italics: distance within species. Distances within species are low, ranging from 0% to 0.68%; distances between species are high, ranging from 7.80% to 13.82%.

## Description of *Plectanocotyle lastovizae* n. sp.

urn:lsid:zoobank.org:act:1BF3310A-3B95-472E-AD39-8289B2D01368

**Description (based on 31 specimens) (Fig. 3; measurements in Table 3).**

Body flattened dorsoventrally. Total length 2,120 (1,100–3,000, $n = 31$), width at level of ovary 310 (200–430, $n = 31$). Haptor symmetrical, with three pairs of pedunculate clamps that measure 125 (90–180, $n = 11$). Each clamp formed by an anterior and posterior jaw; we use the nomenclature of Llewellyn (1956). Anterior jaw composed of a median sclerite 'a' 60 (51–70, $n = 12$) in length which passes to the posterior jaw and two sclerites 'b' 75 (63–85, $n = 12$) in length that maintain the edges of the clamp. Posterior jaw formed of two sclerites 'c' 63 (52–95, $n = 12$) in length shaped as quarter circles. Posterior jaw is also supported by sclerite 'a' which forms an enlargement posteriorly and is followed by two sclerites 'd' 20 (12–26, $n = 5$) in length and 'e' 25 (15–32, $n = 11$) in length. Terminal lappet 265 (160–375, $n = 19$) long and 42 (25–70, $n = 19$) wide, with two pairs of median and lateral hamuli and uncinuli at posterior extremity. Median hamuli 54 (45–60, $n = 12$) long, with little guard and blade; lateral hamuli 54 (42–62, $n = 12$) with a large guard and blade; two uncinuli 9 (5–13, $n = 7$) long.

Mouth subterminal, ventral. Two hemispherical buccal suckers, 34 (25–45) ($n = 15$) in diameter. Median pharynx, small, 48 (38–60, $n = 15$) in length and 42 (30–56, $n = 15$) in width. Short oesophagus. Intestine bifurcate, branches with medial and lateral caeca, extending into posterior region of the haptor, not confluent posteriorly.

Testes 13 (11–15) ($n = 15$) in number in one to two rows. Testes 63 (45–83) ($n = 16$) long and 65 (40–90) ($n = 16$) wide. Vas deferens median, dorsal. Male copulatory organ with eight peripheral slender spines, 112 (90–162) ($n = 19$) long and rwo median spines, 78 (57–90) ($n = 18$) long. Male accessory glands and two correspondent reservoirs one on each side of male copulatory organ.

Ovary tubular, turned back on itself in mid-region of body. Oviduct directed anteriorly. Oötype marked by Mehlis Glands. Vitelline follicles are well developed and located on each

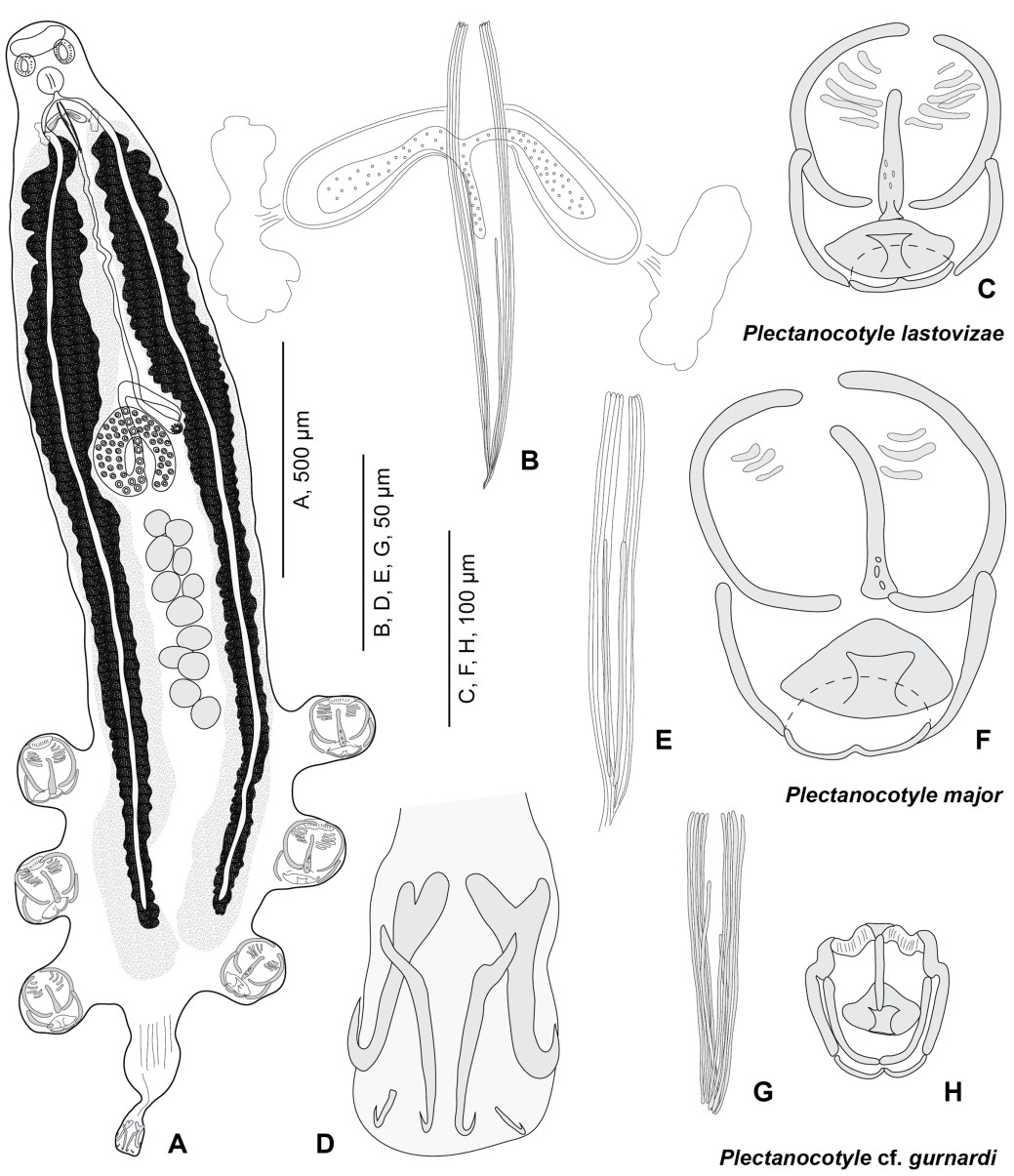

**Figure 3 *Plectanocotyle lastovizae* n. sp. from *Chelidonichthys lastoviza* and comparison with other species.** (A)–(D), *Plectanocotyle lastovizae*, holotype; (E), (F) *Plectanocotyle major*; (G), (H) *Plectanocotyle* cf. *gurnardi*. (A) habitus; (B), (E), (G) spines of the male copulatory organ; (C), (F), (H) clamp; (D) terminal lappet.

side of the body. Uterus ventral and median. Eggs 151 (143–161) (*n* = 10) long, with one posterior polar filament.

**Taxonomic summary**

Synonym: *Plectanocotyle* sp. of Jovelin & Justine, 2001 (*Jovelin & Justine, 2001*)

Type host: *Chelidonichthys lastoviza* (Bonnaterre, 1788)

**Table 3 Measurements of species of *Plectanocotyle*.**

| | *Plectanocotyle lastovizae* n. sp. | *Plectanocotyle* cf. *gurnardi* | *Plectanocotyle major* | *Plectanocotyle major* |
|---|---|---|---|---|
| Hosts | *Chelidonichthys lastoviza* | *Chelidonichthys lastoviza* | *Chelidonichthys obscurus* | *Chelidonichthys obscurus* |
| Localities | Algeria | Algeria | Algeria | Tunisia |
| Source | Present study | Present study | Present study | *Boudaya, Neifar & Euzet, 2006* |
| Body length | 2120 ± 350 (1,100–3,000, $n = 31$) | 2,330 ± 260 (1,900–2,800, $n = 24$) | 4,020 (3,380–4,400, $n = 5$) | 1,841 (1,500–2,300, $n = 11$) |
| Body width | 310 ± 50 (200–430, $n = 31$) | 366 ± 41 (250–430, $n = 24$) | 472 (370–850, $n = 5$) | 442 (210–600, $n = 11$) |
| Buccal organ length | 38 (30–46, $n = 15$) | 41 (29–53, $n = 21$) | 60 (58–62, $n = 4$) | |
| Buccal organ width | 34 (25–45, $n = 15$) | 30 (20–73, $n = 21$) | 59 (55–62, $n = 4$) | 48 (40–60, $n = 22$) |
| Pharynx length | 48 (38–60, $n = 15$) | 60 (50–77, $n = 21$) | 69 (68–70, $n = 4$) | |
| Pharynx width | 42 (30–56, $n = 15$) | 56 (45–70, $n = 21$) | 69 (67–70, $n = 4$) | 45 (30–70, $n = 24$) |
| Clamp length | 125 (90–180, $n = 11$) | 94 (80–103, $n = 10$) | 181 (175–195, $n = 5$) | 137 (90–160, $n = 17$) |
| Clamp width | 101 (35–135, $n = 11$) | 82 (70–90, $n = 8$) | 139 (125–160, $n = 5$) | 110 (85–140, $n = 17$) |
| Sclerites 'a' length | 60 (51–70, $n = 12$) | 53 (42–60, $n = 10$) | 89 (88–90, $n = 5$) | 80 (75–90, $n = 21$) |
| Sclerites 'b' length | 75 (63–85, $n = 12$) | 76 (60–84, $n = 10$) | 127 (123–132, $n = 5$) | 157 (140–180, $n = 21$) |
| Sclerites 'c' length | 63 (52–95, $n = 12$) | 61 (49–76, $n = 10$) | 83 (80–90, $n = 5$) | 72 (65–90, $n = 21$) |
| Sclerites 'd' length | 20 (12–26, $n = 5$) | 27 (20–35, $n = 8$) | 32 (30–35, $n = 5$) | 67 (55–70, $n = 21$) |
| Sclerites 'e' length | 25 (15–32, $n = 11$) | 30 (26–32, $n = 7$) | 28 (27–28, $n = 2$) | 33 (25–40, $n = 21$) |
| Terminal lappet length | 265 (160–375, $n = 19$) | 77 (50–115, $n = 10$) | 61 (56–75, $n = 5$) | 56 (40–80, $n = 13$) |
| Terminal lappet width | 42 (25–70, $n = 19$) | 52 (40–62, $n = 10$) | 40 (36–45, $n = 5$) | 42 (30–50, $n = 13$) |
| Median hamulus length | 54 (45–60, $n = 12$) | 35 (20–51, $n = 10$) | 74 (27–86, $n = 5$) | 32.5 (25–40, $n = 16$) |
| Lateral hamulus length | 54 (42–62, $n = 12$) | 43 (23–53, $n = 10$) | 77 (37–88, $n = 5$) | 33 (30–35, $n = 16$) |
| Postero-lateral uncinulus length | 9 (5–13, $n = 7$) | 11 (6–15, $n = 10$) | 17 (8–20, $n = 5$) | 12 (10–15, $n = 13$) |
| MCO, Peripheral spine length | 112 (90–162, $n = 19$) | 107 (90–125, $n = 11$) | 113 (112–114, $n = 5$) | 107 (90–120, $n = 25$) |
| MCO, Median spine length | 78 (57–90, $n = 18$) | 83 (75–97, $n = 11$) | 75 (73–76, $n = 5$) | 102 (85–110, $n = 21$) |
| Number of testes | 13 (11–15, $n = 15$) | 23 (14–35, $n = 9$) | 20 (19–20, $n = 5$) | 21 (19–22, $n = 11$) |
| Testis length | 63 (45–83, $n = 16$) | 69 (50–115, $n = 8$) | 86 (75–100, $n = 5$) | 57 (30–70, $n = 30$) |
| Testis width | 65 (40–90, $n = 16$) | 68 (45–86, $n = 8$) | 73 (50–86, $n = 5$) | 76 (50–90, $n = 30$) |
| Egg length | 151 (143–161, $n = 10$) | 147 (143–150, $n = 2$) | – | 146 (120–185, $n = 25$) |

**Note:**
Measurements of three species of *Plectanocotyle* collected off Algeria are provided, with a comparison with *P. major* from Tunisia

Type locality: Off Bouharoun, near Alger, Algeria (36°37′24.17″N, 2°39′17.38″E), Mediterranean Sea.

Other locality: off Sète, France, Mediterranean Sea (from similarity of sequences, see discussion).

Microhabitat: gills

Prevalence: 48% (24 infected/50 examined)

Type material: Holotype, MNHN HEL1652; paratypes, 43 slides MNHN HEL1653-HEL1695 (specimens in Canada balsam); vouchers in picrate, 12 slides, MNHN HEL1696-HEL1707; vouchers of specimens used for molecular barcoding, *i.e.*, slides with anterior part of monogeneans, 4 slides, MNHN HEL1723-HEL1726 (Table 1).

Etymology: the species name *lastovizae* refers to the host species, *Chelidonichthys lastoviza*.

Comparative material examined. Note that the number of slides of vouchers of specimens used for molecular barcoding generally exceed the number of sequences (Table 1) because some specimens did not yield usable sequences.

*Plectanocotyle* cf. *gurnardi*. Specimens from *C. lastoviza*, off Bouharoun, Algeria, 12 slides, vouchers deposited in MNHN, MNHN HEL1711-HEL1722; vouchers of specimens used for molecular barcoding, *i.e.*, slides with anterior part of monogeneans, 10 slides including four with sequence, MNHN HEL1723-HEL1726 (Table 1). We do not designate these specimens as *P. gurnardi* (Van Beneden & Hesse, 1863) because their host was different from the type-host (*C. lastoviza* vs *Eutrigla gurnardus*) and their locality was different from the type-locality (Mediterranean vs North Sea). In the absence of molecular information from specimens of *P. gurnardi* from its type-host and type-locality, we provisorily keep the status of this species as pending but remark that it could well be a distinct species. We have recently collected specimens from the type-host to test this hypothesis; this will be the subject of a distinct paper.

*Plectanocotyle major*. Holotype and paratypes from *C. obscurus* off Sète, France, MNHN 268HG—271HG, TJ146—Tj149bis; paratype from off Sfax, Tunisia, MNHN 272HG, Tj150. In all type specimens but one, the length of the MCO spines ranges from 112 to 125 μm, which corresponds to the original description (90–120 μm), but curiously, specimen 271HG from Sète has spines 195 μm in length; this specimen was not in an optimal state and other characters could not be studied. Specimens from *C. obscurus*, off Bouharoun, Algeria, three slides, vouchers deposited in MNHN, MNHN HEL1708–HEL1710. Vouchers of specimens used for molecular barcoding, *i.e.*, slides with anterior part of monogeneans, five slides including three with sequence, MNHN HEL1735–HEL1737 (Table 1).

*Triglicola obscura*. Holotype and paratype of *Plectanocotyloides obscurum* from *Chelidonichthys obscurus*, off Sète, France, MNHN 164TC, Tj178 and Tj179. Vouchers of specimens from *Chelidonichthys obscurus*, off Bouharoun, Algeria, used for molecular barcoding, *i.e.*, slides with anterior part of monogeneans, six slides including three with sequence MNHN HEL1738–HEL1740 (Table 1). For this species, we checked that the length of the spines of the MCO (180–210 μm) corresponded with those of the type specimens (our measurements: 170 μm in both; original description: 175–185 μm). When looking at our voucher slides with only the anterior part, this species can be easily distinguished from the three other species mentioned here by this characteristic alone, which does not overlap (length in all three *Plectanocotyle* species: 90–125 μm).

## DISCUSSION

### Differential diagnosis of *Plectanocotyle lastovizae* n. sp.

*Plectanocotyle lastovizae* n. sp. differs from *P.* cf. *gurnardi* by the morphology and size of clamps (125 × 101 *vs* 94 × 82 µm), the length of terminal lappet (265 × 42 *vs* 77 × 52) and the size of the sclerites of terminal lappet (54 *vs* 43); the most outstanding differential characters are the morphology and size of clamps and the number of testes (13 *vs* 23).

    *Plectanocotyle lastovizae* n. sp. differs from *P. major* by the size of clamps (125 × 101 *vs* 181 × 139 µm), the size of terminal lappet (265 × 42 *vs* 61 × 40), the size of the lateral hamuli of terminal lappet (54 *vs* 77); the most outstanding differential character is the number of testes (13 *vs* 20).

    The three species of *Plectanocotyle* are also distinguished by significant differences in their COI sequences; it is worth mentioning here that our molecular study encompasses all currently known members of the genus (with the exception of *P. gurnardi* if our specimens designated as *P.* cf. *gurnardi* do not actually belong to this species). *Bouguerche et al. (2019a)* compiled results about intraspecific and interspecific variation of COI sequences in polyopisthocotylean monogeneans and found that intraspecific differences ranged 0.2–5.6%; the sequence differences found within each taxon in our study are low enough to be consistent with intraspecific differences, and the differences between taxa exceed the interspecific threshold.

    It is frequent in monogenean taxonomy to distinguish species by measurements of the copulatory organs; we note that this does not apply to the three species of *Plectanocotyle*, which all have spines of the MCO ranging from 90 to 125 µm (with rare measurements up to 162 µm), with overlapping lengths.

### New and other localities for *Plectanocotyle* spp. and *T. obscura*

Since the sequence deposited by *Jovelin & Justine (2001)* as "*Plectanocotyle* sp." from material collected off the Mediterranean coast of France coast off Sète, from the same host, is very similar to our sequences of the Algerian material, it is likely that *P. lastovizae* is present on both the North and South shores of the Mediterranean. Therefore, off Sète, France, is considered an additional locality for *P. lastovizae*, for which the type-locality is off Algeria.

    Sequences from a phylogeny (*Jovelin & Justine, 2001*) produced from the same animals are curiously labelled in GenBank with two names; although designated in the text and GenBank "details" as *Plectanocotyloides obscurum*, they are currently labelled as both *Plectanocotyloides obscurum* (28S, AF311718) and "*Octoplectanocotyla obscura*" (COI, AY009170). The current binomial designation of this species is *Triglicola obscura* (Euzet & Suriano, 1974) Mamaev, 1976. The type-locality of *T. obscura* is off Sète, France; similarity of our COI sequences with this sequence indicates the presence of *T. obscura* off Algeria.

    The present records of *P. major* (type-locality, off Sète, France) and *T. obscura* are first records for Algeria. *Plectanocotyle major* has also been recorded from Tunisia (*Boudaya, Neifar & Euzet, 2006*, *Boudaya et al., 2020*).

There are several records of *P. gurnardi* in the North Sea (*i.e.*, *Køie, 2000*; *Llewellyn, 1941*; *Pugachev & Fagerholm, 1995*); hosts are generally *Eutrigla gurnardus*, with a mention in *Chelidonichthys cuculus* (*Llewellyn, 1941*). In the Mediterranean, the mentions of *P. gurnardi* from its type-host, *E. gurnardus*, off Sète, France are probably valid (*Jovelin & Justine, 2001*; *Tuzet & Ktari, 1971*). In addition, *Plectanocotyle gurnardi* has been recorded from another host, *Chelidonichthys lucerna* (Linnaeus, 1758) (as *Trigla lucerna*) in the Black Sea (*Pogorel'tseva, 1964*) and in the Mediterranean off Turkey (*Akmirza, 2013*). In the absence of molecular information, we do not know if this species from *C. lucerna* is the same as our *P.* cf. *gurnardi* and if it is actually *P. gurnardi*. There is also a mention of *P. gurnardi* on *Chelidonichthys lastoviza* (as *Trigla lineata*) off Greece (*Papoutsoglou, 1975*) which could be either our new species *P. lastovizae* or the species we designated here as *P.* cf. *gurnardi*.

## Nomenclatural status of *Plectanocotyle* and its species

The nomenclatural status of *Plectanocotyle* requires some explanation. *Plectanocotyle elliptica* Diesing, 1850 was described from specimens found on the gills of the White Perch *Morone americana* (as *Labrax mucronatus)*, a Moronidae from North America (*Diesing, 1850*), but was never found again on the same host. *Phyllocotyle gurnardi* Van Beneden & Hesse, 1863 was described from the gills of *Eutrigla gurnardus* (Triglidae) collected off Belgium (*Van Beneden & Hesse, 1863*). *Plectanocotyle lorenzii* Monticelli, 1899 was described from *Trigla* sp. in the Mediterranean Sea (*Monticelli, 1899*), and *Plectanocotyle caudata* Lebour, 1908 was described from *Eutrigla gurnardus* in the North Sea (*Lebour, 1908*). Llewellyn detailed this complex taxonomical situation and concluded (*Llewellyn, 1941*) that *Plectanocotyle gurnardi* (Van Beneden & Hesse, 1863) Llewellyn, 1941 was the valid taxon parasitic on the gills of *Eutrigla gurnardus* (and *Eutrigla cuculus*). Sproston followed that opinion, provided a detailed report of the taxonomic contradictions and redescribed specimens at various states of maturity and contraction (*Sproston, 1946*). Price followed Sproston (*Price, 1961*). Several major authors, including Sproston, Bychowsky and Price (*Bychowsky, 1961*; *Price, 1961*; *Sproston, 1946*) pointed out that *Plectanocotyle elliptica* Diesing, 1850 was never found again and that no *Plectanocotyle* species was ever found again outside the Triglidae. We can therefore consider that nomenclatural issues have been solved in the past and this indicates that species of *Plectanocotyle* are parasitic only on Triglidae. We note, however, that *P. elliptica* is still considered valid in the WoRMS database (*WoRMS, 2021*).

Within *Plectanocotyle*, because of the synonymies proposed by Llewelyn, only two species were recognized before our study (*Llewellyn, 1941*): *Plectanocotyle gurnardi* (Van Beneden & Hesse, 1863) Llewellyn, 1941 and *Plectanocotyle major* Boudaya, Neifar & Euzet, 2006 (*Gibson & Bray, 2001*). However, *Boudaya, Neifar & Euzet (2006)* mentioned that they collected specimens in the Mediterranean Sea from *Chelidonichthys cuculus* (as *Triglus cuculus*) and *Chelidonichthys lastoviza* (as *Trigla lineata*) which belonged to a third species but they did not describe them. *Jovelin & Justine (2001)* using specimens examined by Euzet from off Sète (France, Mediterranean Sea), sequenced partial 28S from *Plectanocotyle gurnardi* from *Eutrigla gurnardus* and *Plectanocotyle* sp. from

 

*Chelidonichthys lastoviza*, and COI from the latter monogenean; unfortunately no COI sequence was published from *P. gurnardi* (*Jovelin & Justine, 2001*).

In this study, we considered that the two *Plectanocotyle* species found on *C. lastoviza* corresponded to a species similar to a known species, *P. gurnardi*, and a new one, *P. lastovizae*. On the basis of host alone, we could have considered the two species from *C. lastoviza* as two new species, but we preferred a conservative approach. We understand that our identification of the species as "*P.* cf. *gurnardi*" should be challenged by a molecular and morphological study of specimens from the type-host and type-locality of *P. gurnardi*.

## Hosts and biology of species of Plectanocotylidae

According to WoRMS (*WoRMS, 2021*), there are 9 valid genera in the Plectanocotylidae. Euzet & Trilles described *Octolabea turchinii* Euzet & Trilles, 1960 and assigned it to the Plectanocotylidae on the basis of the clamp structure (*Euzet & Trilles, 1960*). Yamaguti considered *Octolabea* as the type of his new family Octolabeidae Yamaguti, 1963 (*Yamaguti, 1963*). Later, Euzet & Suriano considered that *Octolabea* should be kept within the Plectanocotylidae, thus rending the Octolabeidae invalid (*Euzet & Suriano, 1973*). In addition, Euzet & Suriano considered that *Octoplectanocotyla* Yamaguti, 1937 was not a member of the Plectanocotylidae because of the structure of its clamps and because its hosts, fishes of the family Trichiuridae, were different from the hosts of other plectanocotylids (*Euzet & Suriano, 1973*). Mamaev and various collaborators created a series of new genera in the 1970's, based on the study of fishes from the Pacific which were not examined by previous authors. Table 4 summarizes the species known and their hosts; it is based on WoRMS (nine genera) with the addition of *Octolabea* and the exclusion of *Plectanocotyloides*, considered a synonym of *Triglicola* according to Mamaev (*Mamaev, 1976*); we provisionally keep *Octoplectanocotyla*, thus reaching a total of 9 genera and 20 valid species.

In the family Plectanocotylidae (Table 4), members of *Plectanocotyle* and *Triglicola* are parasitic of fishes of the family Triglidae (gurnards) and members of *Adenicola, Octolabea* and *Peristedionelia* are parasites of Peristediidae (armoured gurnards); both Triglidae and Peristediidae are Scorpaeniformes. However, members of *Octodiplectanocotyla* are on Trichiuridae (cutlassfishes, Perciformes), members of *Inversocotyle* are on Acropomatidae (lanternbellies, Perciformes), members of *Triglicoloides* are on Chlorophthalmidae (Aulopiformes), and members of *Euzeplectanocotyle* are on Trachichthyidae (Slimeheads, Beryciformes). We do not include the Moronidae within this list since we consider that the record of *P. elliptica* from *Morone americana* (*Diesing, 1850*) was erroneous (*Bychowsky, 1961*; *Price, 1961*; *Sproston, 1946*). Finally, the family Plectanocotylidae includes species parasitic on a wide variety of Teleostei: Perciformes, Scorpaeniformes, Aulopiformes and Beryciformes.

For most of the species described by Mamaev, we found no record in addition to the original description, but *Euzetplectanocotyle hoplosteti* was mentioned again on its type-host *Hoplosteus mediterraneus* in the Mediterranean off Valencia, Spain (*Hernández-Orts, Juan-García & Kuchta, 2016*). Studies on members of *Plectanocotyle*

**Table 4 Species of Plectanocotylidae and their hosts.** Nine genera were recognized in the family Plectanocotylidae Monticelli, 1903 in *WoRMS (2021)* but after Mamaev we consider that *Plectanocotyloides* is a junior synonym of *Triglicola*. We include *Octolabea*, although it was separated from the Plectanocotylidae by *Yamaguti (1963)*. Genera are in alphabetical order. Finally, this table includes nine genera and 20 species.

***Adenicola* Mamaev & Parukhin, 1972.** 1 species.

*Adenicola arabica* Mamaev & Parukhin, 1972 (*Mamaev & Parukhin, 1972b*). Ex *Peristedion adeni* (Peristediidae).

***Euzetplectanocotyle* Mamaev & Tkachuk, 1979.** 1 species.

*Euzetplectanocotyle hoplosteti* Mamaev & Tkachuk, 1979 (*Mamaev & Tkachuk, 1979*). Ex *Hoplosteus mediterraneus* (Trachichthyidae).

***Inversocotyle* Mamaev & Parukhin, 1972.** 1 species.

*Inversocotyle pumilio* Mamaev & Parukhin, 1972 (*Mamaev & Parukhin, 1972a*). Ex *Neoscombrops annutens* (Acropomatidae).

***Octolabea* Euzet & Trilles, 1960.** 1 species.

*Octolabea turchinii* Euzet & Trilles, 1960 (*Euzet & Trilles, 1960*). Ex *Peristedion cataphractum* (Peristediidae).

***Octoplectanocotyla* Yamaguti, 1937.** 3 species.

*Octoplectanocotyla aphanopi* Pascoe, 1987 (*Pascoe, 1987*). Ex *Aphanopus carbo* (Trichiuridae).

*Octoplectanocotyla trichiuri* Yamaguti, 1937 (*Yamaguti, 1937*). Ex *Trichiurus japonicus* (Trichiuridae).

*Octoplectanocotyla travassosi* Carvalho & Luque, 2012 (*Carvalho & Luque, 2012*). Ex *Trichiurus lepturus* (Trichiuridae)

***Peristedionelia* Mamaev & Parukhin, 1972.** 3 species.

*Peristedionelia longisetosa* Mamaev & Parukhin, 1972 (*Mamaev & Parukhin, 1972b*). Ex *Peristedion adeni* (Peristediidae).

*Peristedionelia mosambika* Mamaev & Parukhin, 1972 (*Mamaev & Parukhin, 1972b*). Ex *Peristedion adeni* (Peristediidae).

*Peristedionelia satyrichthysi* Liu & Zhang in Zhang, Yang & Liu, 2001 (*Zhang, Yang & Liu, 2001*). Ex *Satyrichthys rieffeli* (Peristediidae).

***Plectanocotyle* Diesing, 1850.** 3 species.

*Plectanocotyle gurnardi* (Van Beneden & Hesse, 1863) Llewellyn, 1941 (*Llewellyn, 1941*; *Van Beneden & Hesse, 1863*). Ex *Eutrigla gurnardus* (generally designated as *Trigla gurnardus*) (Triglidae); possibly other hosts?

*Plectanocotyle major* Boudaya, Neifar & Euzet, 2006 (*Boudaya, Neifar & Euzet, 2006*). Ex *Chelidonichthys obscurus* (Triglidae).

*Plectanocotyle lastovizae* n. sp. Ex *Chelidonichthys lastoviza* (Triglidae)

***Triglicola* Mamaev & Parukhin, 1972.** 6 species.

*Triglicola australis* Mamaev, 1976. (*Mamaev, 1976*). Ex *Trigla* sp. and *Pterygotrigla picta* (Triglidae).

*Triglicola tonkinensis* Mamaev & Parukhin, 1972 (*Mamaev & Parukhin, 1972a*). Ex *Lepidotrigla* sp. (Triglidae).

*Triglicola dissimmetrica* Mamaev & Parukhin, 1972 (*Mamaev & Parukhin, 1972a*). Ex *Lepidotrigla faurei* (as *Lepidotrigla natalensis)* (Triglidae).

*Triglicola ocellata* Mamaev & Parukhin, 1972 (*Mamaev & Parukhin, 1972a*). Ex *Lepidotrigla* sp. (Triglidae).

*Triglicola ovovivipara* Mamaev & Aleshkina, 1984 (*Mamaev & Aljoshkina, 1984*). Ex *Chelidonichthys hirundo*.

*Triglicola obscura* (Euzet & Suriano, 1974) Mamaev, 1976 (*Euzet & Suriano, 1973*; *Mamaev, 1976*). Ex *Chelidonichthys obscurus* (as *Aspidotrigla obscura)* (Triglidae). Synonyms: *Plectanocotyloides obscurum* Euzet & Suriano, 1974; *Octoplectanocotyla obscurum*, label used in GenBank for sequence originally deposited as *Plectanocotyloides obscurum* (*Jovelin & Justine, 2001*).

***Triglicoloides* Mamaev & Parukhin, 1972.** 1 species.

*Triglicoloides indicus* Mamaev & Parukhin, 1972 (*Mamaev & Parukhin, 1972a*). Ex *Chlorophthalmus agassizi* (Chlorophthalmidae)

which do not pertain to taxonomy are scarce but varied: they included an interpretation of the attachment of the clamps to the gills (*Llewellyn, 1956*), observations on nutrition (*Halton & Jennings, 1965*) and the chemical nature of attachment sclerites (*Lyons, 1966*), a description of the egg, miracidium and hatching (*Whittington & Kearn, 1989*), ultrastructural descriptions of the epidermis (*Lyons, 1972*), clamp sclerites (*Shaw, 1979*)

and spermatozoon (*Tuzet & Ktari, 1971*), and fish stock discrimination (*Boudaya et al., 2020*). A molecular phylogeny (*Jovelin & Justine, 2001*) placed *Plectanocotyle gurnardi* together with *Plectanocotyle* sp. (clearly *P. lastovizae* in view of the similarity of COI sequences) and *Triglicola obscura* in a monophyletic Plectanocotylidae.

### Funding
Travel expenses were funded by the program BIOPARMED- ENVI-MED (https://programmes.insu.cnrs.fr/en/mistrals-en/). Molecular work was funded by MNHN "ATM Barcode" and "ATM PARSUDMED" (www.mnhn.fr). The funders had no role in study design, data collection and analysis, decision to publish, or preparation of the manuscript.

### Grant Disclosures
The following grant information was disclosed by the authors:
BIOPARMED-ENVI-MED.
MNHN "ATM Barcode" and "ATM PARSUDMED".

### Competing Interests
Jean-Lou Justine is an Academic Editor for PeerJ.
The authors declare that they have no other competing interests.

### Author Contributions
- Zouhour El Mouna Ayadi conceived and designed the experiments, performed the experiments, analyzed the data, prepared figures and/or tables, authored or reviewed drafts of the paper, and approved the final draft.
- Fadila Tazerouti conceived and designed the experiments, analyzed the data, authored or reviewed drafts of the paper, and approved the final draft.
- Delphine Gey performed the experiments, analyzed the data, authored or reviewed drafts of the paper, and approved the final draft.
- Jean-Lou Justine conceived and designed the experiments, performed the experiments, analyzed the data, prepared figures and/or tables, authored or reviewed drafts of the paper, and approved the final draft.

### DNA Deposition
The following information was supplied regarding the deposition of DNA sequences:
Fish sequences were deposited in GenBank under accession numbers MG761757–MG761759 and MW788679–MW788687.
Monogenean sequences were deposited in GenBank under accession number MG761760–MG761766 and MW796584–MW796594.

## Data Availability

The raw data are in the Results, Tables 1 & 3, and Fig. 3.

## New Species Registration

The following information was supplied regarding the registration of a newly described species:

Publication LSID: urn:lsid:zoobank.org:pub:B6FB3B21-138D-49AD-B0D6-456AC3F7D36E

*Plectanocotyle lastovizae* LSID:
urn:lsid:zoobank.org:act:1BF3310A-3B95-472E-AD39-8289B2D01368.

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
