# Peer review of "A revision of Plectanocotyle (Monogenea, Plectanocotylidae), with molecular barcoding of three species and the description of a new species from the streaked gurnard Chelidonichthys lastoviza off Algeria"

_PeerJ, doi:10.7717/peerj.12873_

## Round 0.1 · original submission · Major Revisions

I have heard back from two reviewers, both of whom where generally positive about your work. Both have recommended some revisions and corrections, and in my view they seem constructive and fair. I look forward to seeing your revised work.

·

Basic reporting

I believe it is journal policy to include the name of the newly described species in the title. Also, since the article presents much more than a single species description, with a lot of insights in the systematics of Plectanocotyle and the Plectanocotylidae, I would suggest revising the title.

The Introduction session for the moment does not contain a lot of background information (e.g., to have a clearer idea of the previous work on this genus, it would be useful to read already at this point from which host species and which geographical area the two known species have been described). I wonder whether the overview of plectanocotylid species, and the nomenclatural history of Plectanocotyle, would not make more sense as part of the introduction, at least insofar as this is the result of a literature scan rather than of the empirical work contributed in this study. The novel discoveries thanks to the morphological and molecular work in this study can then more easily be put in perspective in subsequent sections of the manuscript.

The text refers a number of times to Adiyodi et al. (2017), but this reference is not in the reference list.

While the English text is generally fine, some formulations strike me as a bit strange, such as (L. 55): “processed for molecules”, (L. 83-84): “using Maximum Likelihood method based on the Tamura 3-parameter”, (L. 99-100) “by DNA barcoding approach”, (L. 100): “present study fish species”, (L. 124): “clamps that measures”, (L. 168): “worthy to mention”

Fig. 1: Can the families be indicated on this figure?

Fig. 2: Since this paper presents an overview of the genus and family, and hence will be very useful to beginners working on Plectanocotyle, I would suggest that the different structures used in the description (posterior and anterior jaw; sclerites a-b-c-d-e; median and lateral hamulus; sclerite types in the male genital apparatus…), or at least those used in the differential diagnosis, be indicated at least once on the figure. This will improve usability to people with no previous experience on this group of monogeneans (like me!).

Table 4: both “Euzetplectanocotyle” and “Euzeplectanocotyle” are used in this manuscript.

Experimental design

L. 42-45: It would be useful to state here that both fish species are gurnards. How were the fish caught and sacrificed?

L. 83-84: Which criterion was used for model selection? What was the proportion of invariable sites?

L. 85-86: Why use a different model in distance calculation and in tree inference? There are certainly reasons thinkable (e.g. comparability with other studies, or with datasets where no model correction was used), but if no such arguments are provided, why not (also) at least provide the distances under the same model as used in tree building?

L. 106: could you explain the discrepancy in length between the amplified fragment for the specimens in the present study and the alignment used for the phylogenetic reconstruction? It is a difference of almost 150 bp, is this because of lack of overlap (other primers) or length differences with published sequences?

Validity of the findings

L. 117-118: It is mentioned that the interspecific distances of over 10% support the species status of the monogeneans under study. Could you comment a bit on this: how does this compare to interspecific divergence over the COI gene in other congeneric (polyopisthocotylean) monogeneans? I know that the amount of existing sequence data for this is not enormous, but I think it is necessary to offer some comparison to have some idea of the cut-offs that the authors use (or propose).

L. 137: The text states 15 as the maximal number of testes, but I think I see 16 of them on Fig. 2?

L. 179-193: This overview of the nomenclatural history of the genus is extremely useful, but I don’t understand everything, could the authors please clarify a bit?
- both 1950 and 1850 are mentioned for Diesing’s work;
- what is the type species of this genus? Is it P. elliptica, but what is the status of this species now? It is only mentioned that the species was never found again and that representatives of this genus are not expected on moronid hosts.
- if I understand it well, P. lorenzii and P. caudata were synonymised with other congeners then? With which ones?

The abstract mentions that the authors “confirm that Plectanocotyloides Euzet & Suriano, 1974 is a junior synonym of Triglicola Mamaev & Parukhin, 1972” but this is not discussed in the text?

Additional comments

I really enjoyed reading this paper. The methodology is sound and thorough, the illustrations are of high quality, and the authors are convincing in arguing why the newly described species indeed represents a distinct species. Importantly, the authors frame the description of this new species in a broader overview of the diversity of the genus and family it belongs to. This overview is both concise and complete, and therefore I think this manuscript will become a handy future reference for scholars working on these marine monogeneans. I commend this approach to a taxonomic study, and this manuscript definitely merits publication. My suggestions only concern the presentation of the paper, where I believe that mainly (1) the methods of the genetic analyses need some more explanation and justification and (2) the taxonomic description and nomenclatural history could use some clarification to be better useable by neophytes.

Reviewer 2 ·

Basic reporting

Some literatures cited in the introduction (line 39) are not relevant with the MS. I don’t know if it is a kind of self citation!

Experimental design

Authors applied rigorous methods to ensure traceability of host and parasite with adequate barcoding but for Plectanocotyle species I suggest to use anterior half (and not the posterior half) for molecular analysis and the posterior half mounted as voucher slide for morphological assessment. The most differential characters are clamps and testes which are located in the posterior half of the body.
Also, I strongly recommend to deposit an hologenophore or a paragenophore (sensu Pleijel et al., 2008). This would have made possible to verify the sequences deposited by Jovelin and Justine (2001) as Octoplectanocotyla obscura and synonymised by the author as Plectanocotyle major.
The reference used for molecular analysis is Ayadi et al. 2017 (and not Adiyodi et al. (2017) (line 62 and line70). I Think that is a typological error

Validity of the findings

The new species Plectanocotyle lastovizae is adequatlilly described. The second species found on the same host Cheilodichthys lastoviza attributed by the authors to Plectanocotyle gurnardi without any comment and discussion. The type host for P. gurnardi is Cheilodichthys gurnadus, one of the authors in a last papers have sequenced partial 28 S from P. gurnardi of Trigla gurnardus (and unfortunately not COI!) it is strongly suggested to have the COI sequence from the type host for comparison

Additional comments

This is an interesting manuscript in a number of respects. This paper is the first attempt to bring order in the genus Plectanocotyle using morphological and molecular analysis. Until now distinction between species is based essentially on the measurement of the clamps which certainly lead to many confusions in the literature (e.g. Llewellyyn, 1941). The table given for the species of Plectanocotylidae and their host can be particularly useful if all synonyms for parasites and host can be added.

Specific comments
- The valid name of the streaked gurnard (as given by worms and Fishbase) is Chelidonichthys lastoviza and not Trigloporus lastoviza. This must be checked throughout the text.
- L 55 : for Plectanocotyle species I suggest to use anterior half (and not the posterior half) for molecular analysis and the posterior half mounted as voucher slide for morphological assessment. The most differential characters are clamps and testes which are located in the posterior half of the body.
- L62 and 70: the reference is Ayadi et al. (2017) and not Adiyodi et al. (2017)
- L80: the reference Olson & Littlewood (2002) or Jovelin & Justine (2001) who made molecular phylogeny is more appropriate here. Phylogeny of Boeger & Kritsky is based on anatomical and ultrastructural characters’
- L 124: the size of the clamps seems to be very variable (90-180). Way only 11 clamps were measured? Is the size of anterior, median and posterior clamps the same? In specimens examined from C. lastoviza in my collection anterior clamps seem to be smaller .
- L125 & 126 the sentence “we use the nomenclature of Llewellyn (1956)” should be moved in the paragraph Nomenclature
- L130 According to Llewellyn (1956) nomenclature used for (e) is the “slender rib-like sclerite in the inner walls of the clamps “and not the posterior most sclerites of the clamps as used by the authors.
- L137. 16 testes is given in the figure of Holotype (Fig 2 A) and (11-15) in the description. this should be verified!
- L143. Add figure for egg if possible
- L153 For comparative material authors should easily examine Paratypes of Plectanocotyle major who is deposited in the MNHN
- L 157 lastoviza with “e”
- L160 The type host of P. gurnardi is Cheilodichthys gurnadus. If they are no type material deposited authors should compare with available bibliography or with specimens of P. gurnardi from Cheilodichthys gurnadus
- Table 3 : measurement is based on carmine stained specimens or Berlese mounted specimens. Standard deviation is not given for all measurement. Prevalence can be added in table 3 for all the parasites
- Table 4 : change Euzeplectanocotyle by Euzetplactanocotyle
- Table 4 : change Peristedionella by Peristedionelia
- Table 4: Another species Octolabea turchinii Euzet & Trilles, 1960 is palced by Euzet & Trilles among the Plectanocotylidae. Yamaguti (1963) establishes new family Octolabeidae Yamaguti, 1963 but Mamaev & Parakhin (1972) replace this species among the Plectanocotylidae. A discussion is needed here.
- Line 314 : change ?? by 6

---

## Round 0.2 · Minor Revisions

I have heard back from two reviewers, who both find your new version well revised. There are some additional helpful comments, but I think these can be easily addressed, and I look forward to seeing a revised version in the near future.

·

Basic reporting

L. 34, end of the “Results” section of the Abstract: I prefer not to use the word “significant” instead of in cases of a formal statistical test.

L. 276-278: this comparison of your results with earlier work on pairwise genetic distance in polyopisthocotylean monogeneans is highly interesting and a strong element in support of your conclusion – but please let the merit of this approach show by mentioning the (approximate?) distances Bouguerche et al. reported intraspecifically and interspecifically.

L. 288-293: at this point it is not clear to the reader what the nomenclatural relationships are between Plectanocotyloides obscurum, Octoplectanocotyla obscura and Triglicola obscura. It is not stated explicitly how the authors assess these names until Table 4 at the end of the text. Please clarify or announce that this will be clarified later on in the paper.

L. 323: minor typo: “issueD”

Figures 1 and 2: please mention in the caption what the scale bar represents.

Experimental design

L. 84-86: why is the voucher slide based on the anterior part? Does this mean the haptor is not necessary for species-level identification with plectanocotylids? It would be good to clarify this, as practices differ strongly between monogenean taxa here. I am wondering because the differential diagnosis, in the Discussion session, contains a lot of information on the haptor, and lines 279-281 seem to suggest that the MCO is not very informative for species-level identification in Plectanocotyle. Which diagnostic feature is hence visible on a voucher slide of the front part of the worm? The testes seem too posterior for that, unless really only the haptor is removed for genetic work?

L. 125: the word “clean” here suggests that something was wrong in the previous alignment matrix. I don’t think that is necessarily the case – incomplete sequences may be a result of low-quality starting material or some hick-up somewhere in the amplification/sequencing process, of course, but can also simply result from e.g. the use of primer sets that amplify regions that only partially overlap. And certainly indels can simply signal meaningful differences between sequences. Did the author consider these indels as problematic or untrustworthy so that they removed these columns in the first matrix? Or did they remove them because of the methodological challenges this poses for downstream analysis? (Though the model-based phylogenetic inference used here could probably handle that in large part anyways.)

L. 165-166: but I thought these indels were removed? Did that not improve the statistical support? (That being said, I also do not find it problematic nor alarming that just over 300 bp of COI sequences do not yield good support values for deeper nodes. This in no way harms answering the questions of this study, I am merely enquiring about how the authors handled indels.)

Validity of the findings

L. 235-243: I appreciate the caution of the authors with respect to characterising specimens when insufficient data is available from the type locality and type host of the species they may belong to. In this case, it is careful, and probably even commendable, to use “Plectanocotyle cf. gurnardi”. However, I do think that doubt should be specified in such cases: e.g. are there morphological differences, of which it is at this point impossible to evaluate whether they represent interspecific or intraspecific variation? Is the original description insufficiently detailed? Or is it because type material is lacking (in which case this should be explicitly stated)? I think the authors should at least make a comparison of what is published about the morphology of P. gurnardi, compare it with their observations they made on the specimens for this study, and assess whether there is reason not to assign them to P. gurnardi. When no potential contradictions arise with the original description, I personally would assign the specimens (supposing they fall within the diagnosis of the described species) to the formally described species. To play the devil’s advocate: why not follow the saying “if it looks like a duck, and quacks like a duck, it’s a duck”? Perhaps that makes me a lumper :-) and I in no way challenge the authors’ decision here; I just would like to see a little bit more justification.

L. 261: on L. 248, the authors mention a type specimen from a representative of Plectanocotyle, P. major, with these spines being 195 µm long. Even though this is only a single specimen, and even though there may be multiple environmental or artefactual reasons for some sort of outlying measurement result, it does make me question whether it is safe to use the length of MCO spines as a highly informative diagnostic trait to distinguish between members of these two genera.

Additional comments

I was already supportive of this MS in its first version, and I was delighted to see that the authors continued this interesting work. Thanks for the opportunity to re-referee this work. I am satisfied with the way the authors incorporated my suggestions, and with the other changes to the MS. However, in view of the study becoming more elaborate now, there are some additional minor things where I think some additional justification or clarification would be needed to render the text palatable for non-Plectanocotyle-specialists. I will detail these below. Actually, I think these are mainly cases where the authors undersell their work or are too cautious; these remarks do not question the soundness of the work. Only under “validity of the findings” I will mention two instances where I would not reach the same conclusion as the authors did based on the current information; I am not necessarily disagreeing so perhaps some sentences in the MS detailing how the authors reached this conclusion would make everything clear.

Reviewer 2 ·

Basic reporting

Compared with the last version, the manuscript is improved but need some revision before publication.
For me, non-native english speakers, English is clear and unambiguous overall, literature is appropriately referenced apart from those in the last paragraph of the introduction (line 64-67) that have absolutely nothing to do with the subject at hand!
Figures and tables are relevant to the content of the article, appropriately described but legend need to be improved:
Figure 1: The phylogenetic tree presented is not of all the members of the Plectanocotyle but based on some sequences from this study and those of Jovelin & Justine (2001). for this figure the legend should be: ML phylogenetic tree constructed based on COI sequences of Plactanocotylidae from this study and the others from Jovelin & Justine (2001).
In the text (line 157) authors Comment NJ tree but in figure they give ML Tree?
Figure 2: It would be more appropriate to say: Molecular phylogenetic analysis of selected sequences of the Plectanocotylidae.
il the end of line 6 of the legend...P. major (90 ML; 88 NJ) and not (80 ML; 88 NJ)
Figure 3: legend: for the male copulatory organ, authors use interchangeably "spine" and "sclerite". I think it's more appropriate to use sclerite. For D change lapped by terminal lapped
for P. cf. gurnardi, It would be advisable to give unfolded claps as for P. major and P. lastovizae
Table 1: the table include 12 new sequences and not 8. I suggest to add sequences for fish with not retrieved monogeneans in the table.
I think there's been some confusion in table 1 the ascension number MG761759 in given for for the same host Br 29 and Monogenea Br 23 MO1!!! please check
Table 2: genetic distances between COI distances of Plectanocotylidae and not of Monogeneans
Table 3: Measurements of Plectocotyle spp. instead of species of Plectanocotyle
In this Table, I suggest to add measurement of Triglicola from Algeria since the authors studied them
Line 33 add congeneric after other

Experimental design

As I mentioned in my last review, posterior half is not appropriate to be in voucher slide for Plectonocotyle spp. because measurements overlap for what remains of the anatomy making traceability difficult

Validity of the findings

line 165-166. Can authors give some references to support this ascertainment
line 187: between Plectanocotyle spp. interspecific distance is ranging between 7.8 and 11.8 % and not 13.8% ( please correct this also in the abstract)
Line 197: the sentence we use the nomenclature of... will be transferred in the M & M
Line 207: in table 3 authors give length and wide for the pharynx but in the description they give diameter!
Line 210: Testes 13 and not 12 as given in table 3
in the taxonomic summary, I suggest to add: synonym: Plectanocotyle sp. of Jovelin and Justine (2001)
line 243: can authors give some others arguments why they suspect that P. cf. gurnardi could be a distinct species?
Line 256 sclerite instead of spine
line 261: Plectanocotyle species: 90-162 instead of 125
line 266: size of the lateral hamuli instead of sclerites
line 281: 162 instead of 125
line 301: Plectanocotyle gurnardi is also recorded from Chelidonichthys cuculus by Llewellyn (1941)
line 327: Lewellyn instead of Llewelyn
le list of

Additional comments

General remarks: please give, in the text, the accepted host name followed in parentheses by the original name used by the authors.

---

## Round 0.3 · Minor Revisions

Thank you for your revision; your work is almost ready to be accepted for publication. Going over the reviewers' comments and your revisions, I have three small points that remain that I wish for you to address, listed below.

1.L. 34, end of the “Results” section of the Abstract: I prefer not to use the word “significant” except for in cases of a formal statistical test.
>>> We looked at various definitions of “significant” in English or American dictionary and this is the right word for what we mean here: “sufficiently great or important to be worthy of attention”.
No change made in the text about this.
Editor: I agree with the reviewer's comment, perhaps "notable" or even a word like "large". While you are correct about the meaning of significant in general English, in scientific papers this often implies statistical analyses, and I would avoid use of the word here.

2. Both reviewers mention:
L. 235-243: I appreciate the caution of the authors with respect to characterising specimens when insufficient data is available from the type locality and type host of the species they may belong to. In this case, it is careful, and probably even commendable, to use “Plectanocotyle cf. gurnardi”. However, I do think that doubt should be specified in such cases: e.g. are there morphological differences, of which it is at this point impossible to evaluate whether they represent interspecific or intraspecific variation? Is the original description insufficiently detailed? Or is it because type material is lacking (in which case this should be explicitly stated)? I think the authors should at least make a comparison of what is published about the morphology of P. gurnardi, compare it with their observations they made on the specimens for this study, and assess whether there is reason not to assign them to P. gurnardi. When no potential contradictions arise with the original description, I personally would assign the specimens (supposing they fall within the diagnosis of the described species) to the formally described species. To play the devil’s advocate: why not follow the saying “if it looks like a duck, and quacks like a duck, it’s a duck”? Perhaps that makes me a lumper :-) and I in no way challenge the authors’ decision here; I just would like to see a little bit more justification.
>>> A simple answer: We have collected specimens of Plectanocotyle gurnardi from the type host and that will be the subject of a whole new paper, now in preparation. It is better to leave the current paper as it is now.
No change was made in the text about this.

And
line 243: can authors give some others arguments why they suspect that P. cf. gurnardi could be a distinct species?
>>> Same answer as for a similar remark from the other reviewer. A simple answer: We have collected specimens of Plectanocotyle gurnardi from the type host and that will be the subject of a whole new paper, now in preparation. It is better to leave the current paper as it is now.
No change was made in the text about this.

Editor: Both reviewers have asked for a little justification, and even if this is another paper, a simple sentence such as "Additional unpublished XXXX data that will be the subject of a future paper support our belief that P. cf. gurnadi is a distinct species" would really be helpful. As this is a taxonomic work, it is best to illuminate your thinking as clearly as possible. I am not asking for a paragraph, but one single sentence even.

3. This last point is up to you, but I do wish you to reconsider this point.
L. 261: on L. 248, the authors mention a type specimen from a representative of Plectanocotyle, P. major, with these spines being 195 µm long. Even though this is only a single specimen, and even though there may be multiple environmental or artefactual reasons for some sort of outlying measurement result, it does make me question whether it is safe to use the length of MCO spines as a highly informative diagnostic trait to distinguish between members of these two genera.
>>> This is an illustration of surprises which arise when you look at specimens deposited in Museums. We hope to provide a better answer to the question of the use of spine length for taxonomy in plectanocotylids when we study more specimens from several species.
No change was made in the text about this.

Editor: Again, here, even a simple sentence or phrase could go a long way towards answering this question.

Overall, while I appreciate your responses and efforts to edit your work, adding a bit more information on these points for future readers will serve your paper well.

I look forward to seeing a revised version of your work.

---

## Round 0.4 · accepted · Accept

I am very pleased to accept this work; thanks for the many years of hard work and effort! Congratulations!